# Chained Risk Assessment for Life-Long Disease Burden of Early Exposures–Demonstration of Concept Using Prenatal Maternal Smoking

**DOI:** 10.3390/ijerph17051472

**Published:** 2020-02-25

**Authors:** Isabell K. Rumrich, Kirsi Vähäkangas, Matti Viluksela, Otto Hänninen

**Affiliations:** 1Department of Public Health Solutions, Finnish Institute for Health and Welfare (THL), P.O. Box 95, 70701 Kuopio, Finland; otto.hanninen@thl.fi; 2Faculty of Science and Forestry, Department of Environmental and Biological Sciences, University of Eastern Finland (UEF), P.O. Box 1627, 70211 Kuopio, Finland; 3Faculty of Health Sciences, School of Pharmacy/Toxicology, University of Eastern Finland (UEF), P.O. Box 1627, 70211 Kuopio, Finland; kirsi.vahakangas@uef.fi (K.V.); matti.viluksela@uef.fi (M.V.); 4Department of Health Security, Finnish Institute for Health and Welfare (THL), P.O. Box 95, 70701 Kuopio, Finland

**Keywords:** Burden of disease, chained risk assessment, developmental origin of health and disease (DOHaD), smoking, pregnancy, low birth weight, preterm birth, overweight, systematic literature review

## Abstract

Traditional risk factors and environmental exposures only explain less than half of the disease burden. The developmental origin of the health and disease (DOHaD) concept proposes that prenatal and early postnatal exposures increase disease susceptibility throughout life. The aim of this work is to demonstrate the application of the DOHaD concept in a chained risk assessment and to provide an estimate of later in life burden of disease related to maternal smoking. We conducted three systematic literature searches for meta-analysis and reviewed the literature reporting meta-analyses of long-term health outcomes associated with maternal smoking and intermediate risk factors (preterm birth, low birth weight, childhood overweight). In the chained model the three selected risk factors explained an additional 2% (34,000 DALY) of the total non-communicable disease burden (1.4 million DALY) in 2017. Being overweight in childhood was the most important risk factor (28,000 DALY). Maternal smoking was directly associated with 170 DALY and indirectly via the three intermediate risk factors 1000 DALY (1200 DALY in total). The results confirm the potential to explain a previously unattributed part of the non-communicable diseases by the DOHAD concept. It is likely that relevant outcomes are missing, resulting in an underestimation of disease burden.

## 1. Introduction

Over the last decades the prevalence of chronic diseases has been increasing. In Finland, about 1.6 million healthy years of life were lost in 2017 due to morbidity and premature mortality [1]. The majority (86%) of this burden was due to non-communicable diseases, while infectious diseases accounted only for 4% [2]. Therefore it is important for the prevention of chronic diseases to identify risk factors. The recent Global Burden of Disease study (GBD2017) analyzed a wide set of risk factors and was able to explain 44% of non-communicable disease burden in Finland in 2017 [2].

The complexity of disease etiology and poorly understood exposure interactions contribute to the lack of understanding of risk factors for disease [3]. The causal risk chain may include a number of independent risk factor or exposure, intermediate states that mediate the effect of the initial exposure onto the outcome and modifying factors that may interact with any factor in the risk chain. Latency times between the initial exposure, intermediate states and the outcome can span from the prenatal phase to decades into adult life (Figure 1) [4]. The above mentioned GBD2017 prevalence-based estimates are mostly cross-sectional, linking directly the exposures for a given target year with the corresponding outcomes. This approach to large extent omits the latency times. 

Already in the 1970 and 80s, significant associations between early conditions and health, disease, and mortality in later life were reported (e.g., [5,6,7,8]) leading to the developmental origin of health and disease (DOHaD) concept. Effects of early exposures during critical windows of development can be persistent and may result in health consequences later in life [9,10,11]. The development of tissues and organ systems during fetal and early postnatal life makes this period especially susceptible for exposure to chemical, physical, and nutritional stress factors. Prenatal short-term, adverse health effects in the fetus can be caused by maternal exposure to environmental chemicals even at doses clearly below the no-effect levels in adults [12]. While early exposure may have adverse long-term consequences, the high developmental plasticity can compensate challenges. The compensation is possible since one genotype can give rise to a range of different physiological or morphological states in response to different environmental conditions during development [5]. With increasing age and the accumulation of risks, the plasticity of the response to challenges decreases and the risk for inadequate responses to new challenge increases potentially resulting in disease onset [13]. 

Prenatal maternal smoking is a known early exposure associated with adverse birth outcomes [14,15]. Despite public health efforts, maternal smoking remains to be prevalent with 14% of all newborns exposed in Finland in 2015 [16]. Exposure to maternal smoking leads to epigenetic changes in the exposed fetus [16,17,18,19], indicating persistent changes in susceptibility to environmental exposures [5]. Emerging evidence is suggesting DNA methylation as a mechanism for the effect of maternal smoking on birth weight [20]. Furthermore, maternal smoking during early pregnancy is causing disproportionate growth restriction, demonstrating differences in susceptibility of organ systems for adverse effects [15]. Growth restriction is evident as early as during the first trimester ultrasound scans [21]. Growth restriction after early exposures is likely attributable to disturbances during organogenesis potentially with persistent effects. Low birth weight (LBW) and preterm birth (PTB) are independent risk factors for later life disease as already demonstrated in the 1980s [6,22,23] and evidence is growing for epigenetic changes associated with these risk factors [24]. Low birth weight and PTB are the most prominent adverse health effects of maternal smoking with epidemiological evidence reaching back to the 1950s [25,26]. Thus maternal smoking is not only causing direct adverse health effects, it also interferes with normal response to environmental challenges increasing the susceptibility for disease. In order to evaluate total health consequences of maternal smoking, these changes in susceptibility should be taken into account as part of the risk chain in disease etiology. 

The aim of this work is to demonstrate the application of the DOHaD concept using maternal smoking as exposure. Specifically to provide chained health risk estimates mediated via selected intermediate risk factors we want to (i) identify relevant health end points in later life; (ii) obtain quantitative estimates for the associations; and (iii) calculate direct and indirect chained population attributable fractions and burden estimates.

## 2. Materials and Methods

Overall, we applied a chained approach to assess health impacts associated with maternal smoking via intermediate risk factors (Figure 2). Three literature reviews were conducted to identify the health effects associated with maternal smoking and the included three intermediate risk factors that have previously been associated with maternal smoking: preterm birth (PTB), low birth weight (LBW), and childhood overweight and obesity. 

Attributable burden of disease was quantified in five groups: first, the direct disease burden associated with maternal smoking, then the burden attributable to the three intermediate risk factors and lastly the fraction of disease burden associated with the three intermediate risk factors that can be indirectly attributed to maternal smoking. 

### 2.1. Identification of Relevant Health Endpoints

The literature reviews with pre-defined inclusion/exclusion criteria were conducted in PubMed (R1–R3 in Figure 2). The searches were limited to meta-analyses to ensure a certain degree of evidence for the association with later life adverse effects. In detail, the inclusion criteria were:Meta-analysis with statistically significant pooled risk estimate;Background disease burden of reported endpoint is available;Free full text is available in English.

The literature search was not restricted in respect to publication year. The searches were conducted in October (Preterm birth and low birth weight, maternal smoking) and November (Childhood overweight and obesity) 2018.

From the initially identified 3766 articles 27 were finally included (Figure 3). The detailed literature search is documented in Appendix A. Relative risk (RR), hazard ratio (HR), or odds ratio (OR) were accepted as risk estimate. If more than one meta-analysis was available for the same endpoint, the newest and largest one was used. Used estimates are listed in Appendix A. The risk estimates identified in the systematic literature reviews were used in the estimation of the population attributable fraction, denoted as RR_k,j_ in Equation (2). A summary of the included studies and risk estimates is listed in Appendix A.

### 2.2. Nested Risk Model

Health impacts were quantified as the burden of disease (BoD) using population attributable fraction (PAF) [27] and background disease burden. Background disease burden of non-communicable diseases was retrieved from Global Burden of Disease (GBD) Results tool [2], including background mortality and morbidity without age weighting and discounting. GBD includes more than 350 mutually exclusive causes (“endpoints”). 

The attributable disease burden (*AB*) was calculated for maternal smoking and each risk factor (identified by subscripts *k*) by combining population attributable fraction (*PAF*) with background disease burden (*BoD*) and summed over all endpoints (*j*) (Equation (1); Equation (2)).
(1)$$AB_{k,j}=\;\underset{j}{\sum }(PAF_{k,j}\times \;BoD_{j})$$

Population attributable fraction was derived from the fraction of exposed population in the whole target population (*f*) and the associated relative risk (*RR*_*k*,*j*_) (Equation (2)). Parameter values for *RR*_*k*,*j*_ are listed in Appendix A. For each exposure-response association identified in the systematic literature reviews, a separate *PAF* was estimated.
(2)$$PAF_{k,j}=\;\frac{f_{i}\left(RR_{k,j}-1\right)}{1+f_{i}\left(RR_{k,j}-1\right)}$$

In a chained assessment the indirect impact of maternal smoking mediated via the three selected intermediate risk factors was quantified. The contribution of maternal smoking to disease burden attributable to the intermediate risk factors (*AB_i_*) was estimated using *PAF* approach and the disease burden attributable to each intermediate risk factor (Equation (3)). *PAF* is calculated as in Equation (2) with the special case of *k* being maternal smoking and *j* the intermediate risk factor (see Table 1 for parameter values).
(3)$$AB_{i}=\;\underset{j}{\sum }\left(PAF_{k,j}*\;AB_{j}\right)$$

The MATEX cohort was used for exposure data [16]. In Finland in 2016, about 6.8% of newborns were exposed to smoking only during the fist trimester and additional 7.0% were exposed to maternal smoking also after the 1st trimester (Table 1). Statistically significant association for smoking during early pregnancy was reported only for low birth weight. For preterm birth and childhood overweight and obesity only smoking throughout pregnancy was considered a relevant exposure. 

The prevalence of PTB and LBW with corresponding risk estimates for the associations with maternal smoking were identified from the MATEX birth cohort [15]. The prevalence of childhood overweight and obesity were estimated from Finnish population growth curves based on the WHO definition of overweight and obesity (Table 1) [28]. PAF for LBW and PTB have been separately calculated for smoking only during early pregnancy and smoking in late pregnancy to take into account differences in risk estimates.

## 3. Results

Three reviews were conducted to identify relevant endpoints and risk estimates to quantify disease burden associated with maternal smoking and intermediate risk factors in a chained risk model. In total, 27 meta-analyses were identified as eligible, resulting in 24 diseases to be included in the risk estimation (Table 2). The health outcomes included cancer, cardiovascular diseases, mental and cognitive diseases, diabetes, asthma, congenital anomalies, and chronic kidney disease. 

The ORs identified in the systematic literature reviews ranged from 1.09 (95% CI 1.02–1.17) for the association of PTB with acute leukemia and maternal smoking with nervous system cancers, to 5.6 (95% CI 2.34–13.3) for childhood obesity with diabetes later in life. The only other ORs above 2 were for PTB and attention deficit/hyperactivity disorder (OR 2.6, 95% CI 1.85–3.78), childhood overweight and diabetes (OR 2.4, 95% CI 1.54–3.75), and PTB and intellectual disability (OR 2.03, 95% CI 1.79–2.31) (see the Appendix A). The average population attributable fraction was 2.5% with individual PAFs ranging between 0.4% (PTB and depression, LBW and depression, PTB and acute leukemia) and 16% (childhood overweight and diabetes) (see the Appendix A).

The overall burden of non-communicable diseases in Finland in 2017 was 1.4 million DALY [2]. The disease burden attributable to endpoints associated with maternal smoking or intermediate risk factors included in this work was about 380,000DALY (28% of the total non-communicable disease burden) (Table 2). Cardiovascular disease burden (242,000 DALY) was the main contributing factor to the included burden, followed by diabetes (56,000 DALY) and cancer (44,000 DALY). Depression was the smallest contributor to the total disease burden with about 300 DALY.

Out of 1.4 million DALY caused by non-communicable diseases, about 33,500 DALY could be directly attributed to intermediate risk factors included in this work. Childhood overweight was associated with the greatest health impact (28,000 DALY), followed by LBW (4000 DALY), PTB (1400 DALY), and the direct impact of maternal smoking (170 DALY) (Table 2; Figure 4).

The contribution of maternal smoking to the disease burden of the three intermediate risk factors (childhood overweight, LBW, PTB) was about 3% (1040 DALY) (Figure 4). In total, 0.1% of the total non-communicable disease burden was attributable to maternal smoking in the chained impact assessment. Congenital anomalies (115 DALY) were associated with the highest disease burden directly attributable to maternal smoking, followed by asthma (39 DALY).

Disease burden due to diabetes and cardiovascular diseases were the main contributors to the attributable burden with a total of 18,000 DALY and 12,000 DALY respectively (Figure 5), corresponding to 550 DALY and 370 DALY indirectly attributable to maternal smoking. The indirect contribution of maternal smoking to disease burden exceeds the directly attributable disease burden (Figure 5).

Overall, the risk factor attributable morbidity (years lived with disability; YLD) was slightly larger (55%) than the health loss due to premature mortality (years of life lost; YLD) (45%). The disease burden directly attributable to maternal smoking and PTB was more clearly dominated by YLD, 77% and 82% respectively, while YLL dominated the disease burden attributable to LBW with 60% (Figure 6). The proportions of YLL and YLD are similar for the direct disease burden attributable to intermediate risk factors and the disease burden indirectly attributable to maternal smoking.

## 4. Discussion

As the relative rate in prevalence of non-communicable diseases is rising due to better control of infectious diseases and an aging society, the scientific and medical community is aiming to identify modifiable risk factors in disease etiology, which are needed to design efficient strategies for disease prevention. To date, the established risk factors fail to explain a majority of disease burden. Increasing understanding of epigenetics and the mixture of internal and external exposures underline the complexity of disease etiology and the subtle sub-clinical effects of many exposures throughout the life course, which as a sum affect the susceptibility for disease. The developmental origin of health and disease (DOHaD) suggests pre- and early postnatal exposures as especially promising for understanding later life susceptibility. The health risks of developmental exposures are not yet well understood for the whole life course. In this work we demonstrated the potential of developmental exposures to explain disease burden throughout the life course. 

We demonstrate that the DOHaD principle can be used to characterize health consequences of developmental exposures throughout the life course. We estimated the health effects of maternal smoking, a well characterized prenatal exposure and risk factor, using a chained risk model. The chained assessment accounts for health effects of risk factors intermediate in the sequential chain from maternal smoking to chronic diseases: PTB, LBW, and childhood overweight and obesity. However, the risk characterization is limited by data gaps. Firstly, the GBD background health data are not available for early markers of disease or non-medical endpoints, such as lower academic achievements or generally decreased quality of life. The assessment is further complicated by the availability of some health effects, such as high cholesterol levels or markers of metabolic disease as risk factors in the GBD database. These risk factors cannot easily be used for the estimation of DOHaD attributable disease burden due to potential double-counting of disease burden. Secondly, the chained risk model was limited by the availability of exposure-response relationship between maternal smoking or intermediate risk factors and later life effects with high evidence for causality. A systematic approach is needed to evaluate the evidence for causality and strength of association between early exposures and health in later life, which then can be used in a comprehensive risk assessment. Thirdly, the burden of disease is estimated in a cross-sectional design. In the risk assessment of prenatal and early postnatal exposures the time lag between exposure and outcome is potentially decades. A lifetable approach, modelling the changes in exposure level and background burden over the life course, is needed to gain more detailed insight into the distribution of associated disease burden. In general, the results of this scoping work to demonstrate the concept must be validated and refined by systematic bottom-up analyses of exposures, risks and long term disease burden trends.

The health effects quantified in this work is likely to be only the tip of the iceberg. We excluded studies that reported only at all cancers aggregated (“Any cancer”) due to clear differences in etiology of cancer types casting doubt on the reliability of such a grouping. However, fairly consistent increases in risk for certain cancer types of around 20% have been reported for childhood overweight [30], possibly contributing considerably to the total attributable disease burden. A wide range of health effects, from sub-clinical metabolic changes to mental and cognitive affecting later life happiness and productivity, were excluded from the risk model due to a lack of reliable background disease burden (Table 3). GBD2017 estimates suggest a total disease burden attributable to metabolic risk factors of about 383,000 DALY in Finland in 2017, accounting for roughly half of the total burden attributable to risk factors included in the GBD2017 study. High BMI and high fasting plasma glucose level are the main contributors to metabolic risks with about 160,000 attributable DALY each, followed by high cholesterol levels (82,000 DALY) [31]. Considering that LBW has been shown to be associated with metabolic syndrome [32], the estimates of attributable disease burden presented in this work are likely to underestimate the true disease burden by excluding early stages of diseases as they are only available as risk factor and not cause in the GBD database. Neonatal preterm birth is included as a cause for disease burden in the GBD2017 study, accounting for 7026 DALY in Finland in 2017 [33]. This is considerably higher than the roughly 1300 DALY we estimated as attributable to PTB. In the estimation of PTB disease burden the GBD2017 include infectious diseases and a wide variety of severe long term effects, such as blindness, cognitive and motor impairments, potentially contributing to the higher attributable disease burden [34]. 

Furthermore, our results may underestimate the disease burden because trans-generational health effects in the offspring of the in utero exposed population are not accounted for. Asthma [35] and gestational diabetes [36] are examples of the health effects associated with grand-maternal smoking. Additionally, we did not take increased risk for all natural cause premature mortality into account. Caution is advised that the BoD estimates presented in this work are order of magnitude estimates and will change with accumulating data. LBW and PTB are partly overlapping conditions with LBW (birth weight < 2500 g) being more prevalent in preterm infants than term infants. Thus, the associated disease burdens may be overlapping. 

Up to 0.1% of the total non-communicable disease burden was attributable to maternal smoking. The total share of included early life risk factors was associated with up to 3% of the total burden, demonstrating the potentially devastating impact on health decades later. Although the chained risk model is based on an extensive literature review to identify health outcomes of the risk factors, the risk assessment was constrained by heavy data limitations as pointed out above. Thus we expect the actual attributable disease burden to be substantially higher. 

Roughly 34,000 DALY were attributable to developmental origin in this work. Thus burden of developmental risk factors is comparable with disease burden attributable to air pollution in Finland. In 2015 about 35,000 DALY were attributable to particulate matter (PM2.5, PM10), nitrogen dioxide, and ozone [58]. The health impact of ambient ozone has been estimated to be between 750 and 1500 DALY [2,58], ranging in the same order of magnitude as the total health impact of maternal smoking (1200 DALY) estimated in this work. Air pollution is recognized as a serious public health problem with high priority in research and policy making [59]. This work highlights the importance of the developmental period for overall health for the rest of the life. Providing the best start to a healthy life requires multidisciplinary efforts to reduce any harmful insults during sensitive developmental periods by improving awareness of the parents and families of harmful exposures such as smoking and contaminants in the ambient environment. The identification of unborn babies at risk and the management of these pregnancies should be public health priority. This includes efforts to educate the expecting families about risk factors and harmful exposures such as (second hand) tobacco smoking, unbalanced diet and stress. 

While this work is based on estimates for the Finnish population, maternal smoking is a global public health problem [60]. Recent rates of maternal smoking remain at around 10% in the Nordic Countries and Australia [61,62] and up to 38% in Ireland [60]. Although population attributable fractions and background burdens vary between countries, the order of magnitude of the relative importance of diseases in the total burden in Western countries is comparable. Smoking during pregnancy and risk factors during early life are likely to contribute to disease burden in a similar relative order of magnitude as in Finland. Similarly, LBW, PTB, and childhood overweight are public health concerns globally with high prevalence rates [63,64,65]. Thus legislative smoking bans, as introduced in Finland with the latest revision of the Tobacco act, will considerably improve public health. Disease burden in active smokers will greatly decrease, but also in prenatally exposed children. The associated decrease in risk factors for later life disease will have long-term beneficial effect on chronic disease burden in the population.

### 4.1. Biological Plausibility

Application of epigenetics and exposome methods are in their infancy in environmental epidemiology. As there is not sufficient understanding of disease etiology to link maternal smoking associated epigenetic changes, exposome, and sub-clinical effects to background disease burden, we propose that epigenetic changes caused by maternal smoking are reflected in the sequential, intermediate risk factors [16,19,24]. Therefore, a chained risk assessment from maternal smoking via these mediators to health endpoints is a promising approach, even though there is limited epidemiological evidence for a direct association between maternal smoking and the health endpoints. 

The nested risk model relies on the assumption of causality between the risk factors and included health outcomes. By including only endpoints identified as statistically significant in well conducted meta-analyses we hope to ensure a reasonable level of reliability. However, statistical significance in meta-analyses alone is not sufficient to infer causality, especially with long lag time between the exposure of concern and health endpoint and considerable potential for confounding [66]. Mechanistic explanations are slowly emerging for the increased susceptibility to diseases later in life, due to sub-optimal developmental environment. It has been suggested that at least (i) alterations in gene expression, potentially altering detoxification of xenobiotics, mediated via epigenetic changes; (ii) changes in neuroendocrine system, including increased levels of glucocorticoids and dysregulated hypothalamic-adrenal axis; (iii) irreversible changes in organ structure; and (iv) cellular aging are possible mechanisms explaining an increase of the observed effect [67].

Currently, comprehensive health impact assessment of developmental exposures along the life course is complicated by the limited availability of birth cohorts followed long-term. Such cohorts are necessary to better understand the disease etiology with potentially interacting risk factors. Advanced, careful study design, and sufficient study sizes are essential for causal inference, especially with potentially decade long latency [66]. Register-based study design can facilitate the needed big study sizes and it reduces the need to contact study participants for follow-up over long time periods. Additionally, the majority of epidemiological studies include LBW as in indicator for sub-optimal in utero development, despite the evidence that general LBW is not a good predictor for future health. Growth rate, gestational age at birth and body composition and proportions need to be taken into account to distinguish pathologically small infants from constitutionally small infants [68]. There is already sufficient evidence for differences in etiology and long term consequences of the so-called symmetrical and asymmetrical growth restricted newborns. Symmetrical growth restriction is associated with a poorer prognosis, reduced cell number, proportional reduction in all biometrical indices, and the period of insult is in earlier gestation [69]. 

### 4.2. Uncertainties

Health effects were quantified on cross-sectional data for 2017 leading to all health effects of maternal smoking attributed to background disease burden of the same year, even if the endpoint is in the elderly and the exposure was theoretically decades ago. Lack of sufficient exposure, risk, and background health data prevents a life table approach linking exposure rates with the corresponding outcome decades later. However, over the last 25 years maternal smoking during early pregnancy rates have been fairly stable around 15% in Finland. Therefore, we do not anticipate that changes in smoking patterns and other disease risk factors affect the order of magnitude of the burden of disease estimates.

The burden of disease estimates are directly dependent on estimates of exposure and risk. In this work we employed a priori selection criteria to include endpoints and risk estimates to aim at a consistent level of evidence. Nevertheless, the used meta-analyses and with that our disease burden estimates are subject to uncertainties. For most associations included in this work, causality between maternal smoking and the outcomes is not proven. Nevertheless, lack of available meta-analysis may not necessary suggest lack causality, but may only indicate that there are not sufficient number of studies, yet, to conduct a meta-analysis or conclude on causality. In addition, some meta-analyses may have been missed in the literature search and review. Meta-analyses themselves are sensitive to bias, such as publication bias, selection bias and bias in the original studies included in the pooled analyses. Additionally, uncertainties arise from the low specificity of LBW as endpoints. Potentially, there are different subgroups of LBW, such as in term or preterm newborns and symmetrically or asymmetrically growth restricted newborns. Each of these subgroups may be associated with a distinct group of later life consequences and some may be associated with maternal smoking or nicotine use, while others are not. Currently, there is limited evidence for clear differences between subgroups and more studies distinguishing these subgroups are needed, especially focusing on long-term health effects. Furthermore, there is a considerable overlap in LBW and PTB, i.e., a big fraction of prematurely born children is born with a birth weight below 2500 g. Published studies do not always clearly distinguish between the both and attributed long term effects may not be strictly associated with either of them, but more generally with immaturity at birth. In epidemiological studies the follow up time is limited by the availability of study populations with sufficient information on prenatal exposures. Epidemiological studies with a life-long follow up are needed to analyze any possible health consequences, which may be missed by short follow up or small study population. 

Most risk estimates included in this work are reported for specific age groups, which may be chosen out of convenience and not for scientific reasons. We apply the risk estimates partly for wider age ranges than originally reported. This may lead to overestimation of the attributable disease burden, if the association was not valid outside the reported age range. Similarly, there may be a certain degree of mismatch between endpoints reported in meta-analyses and corresponding background disease burden endpoints. We do not anticipate that any mismatch would be that substantial, that it would change the reported attributable burden of disease.

## 5. Conclusions

The developmental origin of disease approach was shown to be applicable in a chained health risk assessment. Tentative estimates for the life-long health impacts in of maternal smoking, mediated via established intermediate risk factors at birth and during childhood, were obtained. Burden of disease attributable directly to maternal smoking and associated risk factors for disease later in life was quantified. The contribution of maternal smoking to the burden of disease attributable to the risk factors was estimated as indirect burden in a chained risk assessment. Maternal smoking was associated with 24 endpoints and a corresponding disease burden of 1211 DALY. The total disease burden attributable to preterm birth, low birth weight and childhood overweight and obesity was 33,500 DALY. 

Substantial part of the overall burden of disease due to chronic diseases has not yet been associated with any risk factors. Chained risk analysis based on developmental exposures is one of the most promising approaches to tackle the unexplained burden. Although the direct health impact of maternal smoking is small, it is associated with a range of risk factors for non-communicable diseases. Taking this risk chain into account emphasizes the high susceptibility for disease later in life caused by developmental exposures. Later in life effects should be considered in comprehensive risk assessment of exposures in early life.

Developmental origin of disease stresses the importance of a safe and healthy environment during developmental periods to achieve healthier life. Consolidated efforts are needed to educate expectant parents about life long consequences of sub-optimal environment during development of their child. Better understanding of life-long consequences of prenatal and early postnatal exposures is needed. The existing birth cohorts with detailed information about prenatal and early postnatal life did mostly not reach elderly age yet, and therefore epidemiological studies of the DOHaD paradigm are complicated and need to continue. 

## Figures and Tables

**Figure 1 ijerph-17-01472-f001:**
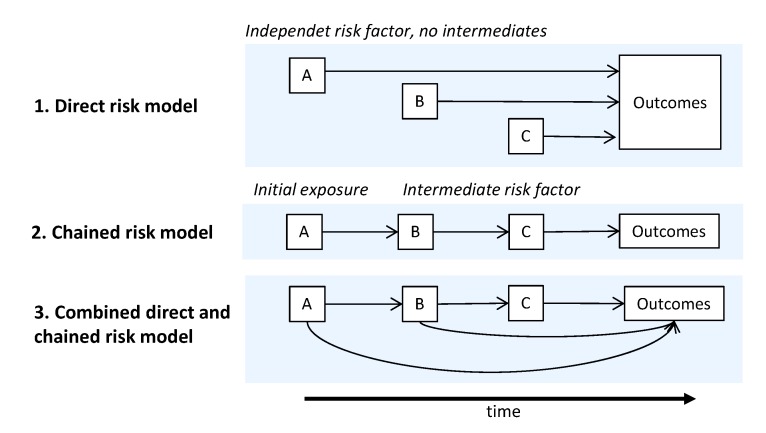
Causal models for disease etiology describing interactions between risk factors and the outcomes (modified from [5]). The outcomes can be immediate, or occur later in the life. A, B, C as generic placeholders for risk factors and outcomes in the causal models.

**Figure 2 ijerph-17-01472-f002:**
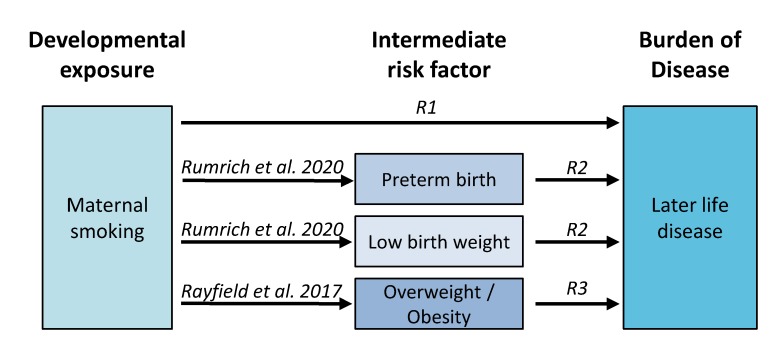
Design of the chained risk model for maternal smoking. Review 1 (R1) identifies later life health outcomes associated directly with maternal smoking and reviews 2–3 (R2, R3) those mediated via the three intermediate factors.

**Figure 3 ijerph-17-01472-f003:**
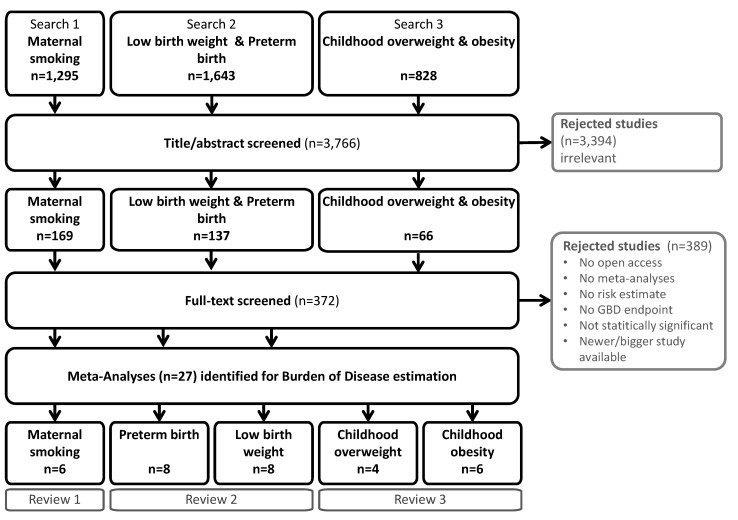
Systematic umbrella literature searches processes for reviews 1–3.

**Figure 4 ijerph-17-01472-f004:**
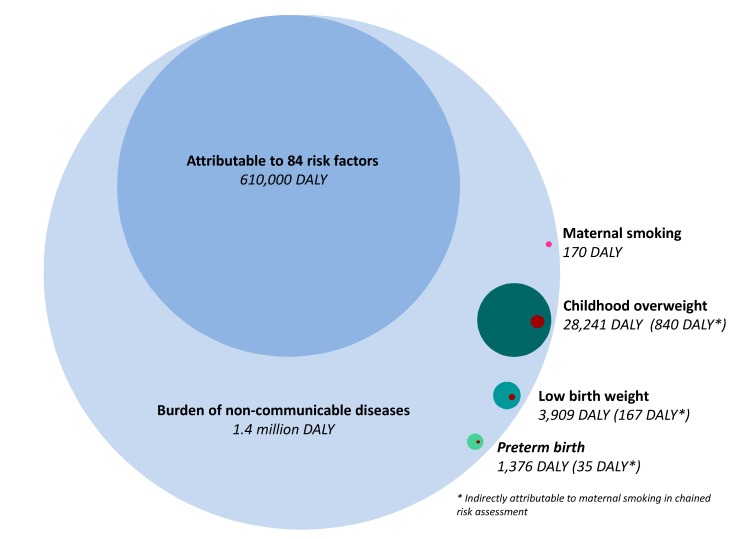
Bubble plot indicating the total burden of disease in Finland in light blue (2017) with the attributable fractions due to risk factors included in the Global Burden of Disease Study (blue), and maternal smoking (direct burden in pink, indirect burden in dark red) and intermediate risk factors considered in this work (green shades). Based on data from [2].

**Figure 5 ijerph-17-01472-f005:**
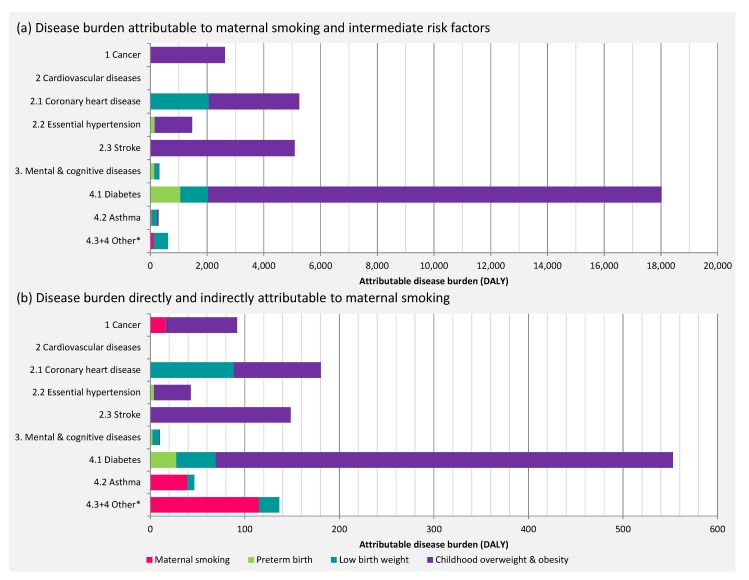
Attributable disease burden (34,000 DALY) by endpoint: (**a**) direct impact; (**b**) direct and indirect impact of maternal smoking (1200 DALY).

**Figure 6 ijerph-17-01472-f006:**
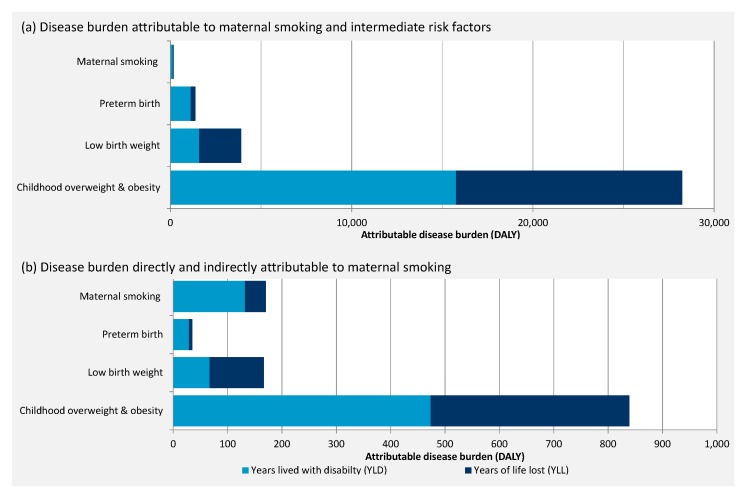
Maternal smoking and risk factor attributable disease burden divided by years lived with disease (YLD) and years lost due to premature death (YLL); (**a**) direct impact; (**b**) direct and indirect impact of maternal smoking.

**Table 1 ijerph-17-01472-t001:** Risk factor definition and population attributable fraction (PAF) estimation for the association between maternal smoking and the intermediate risk factors in this work.

Population Attributable Fraction Input	Maternal Smoking	Preterm Birth (PTB)	Low Birth Weight (LBW)	Childhood Overweight and Obesity (12–18 Years)	
				Overweight	Obesity
Definition	Tobacco smoking	Gestational age < 37 weeks	Birth weight < 2500 g	BMI ≥ 25 at	BMI ≥ 30
**Exposure** only during 1st trimester throughout pregnancy	6.8 % ^a^ [16] 7.0 % ^b^ [16]	n/a [15] 4 %[16]	3 % [16] 3 % [16]	n/a 14% [28]	n/a 3% [28]
Risk estimate (RR)		1.38 [15]	1.11 ^a^ [15] 2.22 ^b^ [15]	1.37 [29]	1.55 [29]
**Population attributable fraction** only during 1st trimester throughout pregnancy		n/a 0.03 ^b^	0.01 ^a^ 0.08 ^b^	n/a 0.03 ^b^	n/a 0.04 ^b^

n/a no statistically significant association, ^a^ Estimate for smoking only during 1st trimester, ^b^ Estimate smoking: smoking throughout pregnancy.

**Table 2 ijerph-17-01472-t002:** Summary of identified endpoints in three umbrella systematic literature reviews and corresponding attributable burden of disease (in DALY).

Endpoint		Age Group	Background Burden (in DALY)	Attributable Disease Burden (in DALY)				
				Maternal Smoking	Preterm Birth	Low Birth Weight	Childhood Overweight & Obesity	Total
1	Cancer							
1.1	Acute leukemia	<20 years	806	3.9	6.8	n/a	32	43
1.2	Lymphoma	<20 years	238	7	n/a	n/a	n/a	7
1.3	Nervous system cancer	<20 years	832	5.2	n/a	n/a	n/a	5.2
1.4	Testicular cancer	<20 years	11	n/a	n/a	0.22	n/a	0.22
1.5	Wilms’ tumor	<20 years	95	n/a	1.6	n/a	n/a	1.6
1.6	Other cancer ^a^	≥20 years	41,647	n/a	n/a	n/a	2570	2570
2	Cardiovascular							
2.1	Coronary heart disease	All ages	157,588	n/a	n/a	2066	3184	5250
2.2	Essential hypertension	All ages	12,532	n/a	153	n/a	1323	1477
2.3	Stroke	All ages	71,952	n/a	n/a	n/a	5088	5088
3	Mental & cognitive disorders							
3.1	Depression	≥20 years	292	n/a	n/a	2.6	3.5	6.1
3.2	Autism	All ages	4528	n/a	55	167	n/a	223
3.3	Attention Deficit/Hyperactivity Disorder	All ages	703	n/a	43	n/a	4.2	47
3.4	Intellectual disability	All ages	1059	n/a	42	n/a	n/a	42
4	Other groups							
4.1	Diabetes	All ages	55,898	n/a	1059	977	15,987	18,023
4.2	Asthma	All ages	13,809	39	14	189	47	290
4.3	Congenital anomalies ^b^	All ages	8593	115	n/a	n/a	n/a	115
4.4	Chronic kidney disease	All ages	11,807	n/a	n/a	506	n/a	506
	Sum		382,388	170	1376	3909	28,241	33,696
	Indirectly attributable to maternal smoking			170	35	167	839	1211
	Indirectly attributable to maternal smoking (%)				3 %	4 %	3 %	4 %

n/a: no statistically significant association identified. ^a^ Including esophageal adenocarcinoma, liver cancer, multiple myeloma, pancreatic cancer, thyroid cancer. ^b^ Including the following congenital anomalies: heart anomalies, orofacial clefts, digestive anomalies, musculoskeletal anomalies.

**Table 3 ijerph-17-01472-t003:** Exemplary outcomes associated with maternal smoking and associated intermediate risk factors for which no background burden was available and therefore not quantified in this work *.

Health Effect	Maternal Smoking	Risk Factor		
		Preterm Birth	Low Birth Weight	Childhood Overweight/Obesity
**Cardiovascular diseases**				
Atherosclerosis				[37]
**Cognitive and Mental**				
Pediatric Quality of Life Inventory Index				[38]
Academic performance		[39]		
Higher education qualification		[40]		
Intelligence			[41]	
Criminal/Deviant behaviour	[42]			
Social benefits		[40]		
**Respiratory**				
Wheezing	[43,44]	[45]	[45,46]	
Decreased lung function (FEF75, FEV1, FVC)		[47]	[47,48]	
**Metabolic**				
Cortisol secretion	[49]		[50]	
Metabolic syndrome			[32]	[51]
Total cholesterol			[52]	[53]
**Other**				
Congenital anomalies	[54]			
Age at menarche	[55]			
Bone mineral density				[56]
Periodontal disease				[57]

* All associations listed here were statistically significant at 95% confidence in the reviewed meta-analyses.

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
