# Peer review of "Chained Risk Assessment for Life-Long Disease Burden of Early Exposures–Demonstration of Concept Using Prenatal Maternal Smoking"

_ijerph, 2020, doi:10.3390/ijerph17051472_

Round 1

Reviewer 1 Report

Important national issue for Finland. Perhaps include in title this is an analyses of three systematic reviews. 
perhsps consider impact legislative smoking bans on maternal smoking.

what other limitations exist using these methods of analysis on all study types. 

Author Response

Dear Reviewer,

Thank you for your time and effort and the highly useful comments. We have taken them into account and adjusted the manuscript accordingly. Please find below our detailed responses. Additions and changes are marked in the manuscript with Word Track changes.

  1. Important national issue for Finland. Perhaps include in title this is an analyses of three systematic reviews

We appreciate your suggestion to mention systematic reviews in the title. To highlight that the work is based on systematic reviews we added “systematic literature review” in the key words. Additionally we expanded the section in the abstract in lines 22f. in the revised manuscript .

  1. Perhaps consider impact legislative smoking bans on maternal smoking.

This is a very valuable suggestion. We fully agree that the impact of legislative smoking bans should be considered in our manuscript.. Thus we added a paragraph discussing the expected changes. Please see lines 331ff. in the revised manuscript.

  1. What other limitations exist using these methods of analysis on all study types. 

We agree that systematic literature review and burden of disease assessment methods have a variety of limitations, as elaborated throughout the discussion section We expanded this section to discuss the limitations in more detail, e.g. lines 388f. and lines 400ff. in the revised manuscript.  In this work we assume the validity of background burden of disease estimated published by the Global Burden of Disease Study collaborators. Ideally, this should be validated by independent assessment in Finland.

As our work is based on a literature of meta-analysis, it is also sensitive to any potential bias and error in the original studies, meta-analyses and literature searches (e.g. exposure misclassification, recall bias, publication bias). At the same time our work highlights the limitations of data availability and in reporting in the original studies. In some cases no clear differentiation between low birth weight and preterm birth was made, assuming they are similar risk factors with similar later life consequences. However, future studies need to consider at least these two groups separately. It would be better to analyse better defined subgroups (e.g. small for gestational age, term low birth weight, preterm birth with normal/low birth weight)  to gain a better understanding of these risk groups.

As discussed in the article (lines 260ff. and 400ff. in the revised manuscript), this work was limited by the availability of original studies and meta-analysis. This underlines the importance of continued research with robust study design and outcome definitions, as well as good reporting and publications of the studies, even if they are null-results.

Reviewer 2 Report

Congratulations for this article.

Please explain with more detail how you used values from systematic review. Please provide a table showing the articles used  for the several estimations.

Table 2 is not crear. Please indicate what represents the numbers under "risk factors"  

Author Response

Dear Reviewer,

Thank you for your time and effort and the highly useful comments. We have taken them into account and adjusted the manuscript accordingly. Please find below our detailed responses. Additions and changes are marked in the manuscript with Word Track changes.

Congratulations for this article.

  1. Please explain with more detail how you used values from systematic review. Please provide a table showing the articles used  for the several estimations.

Thank you for your valuable comments. We clarified the use of the risk estimates from the systematic reviews in the estimation of the attributable burden in lines 122ff. and 143f. in the revised manuscript. Additionally we added a reference to Table S1 in the supplemental material. The table lists all endpoints, their association with maternal smoking and/or the intermediate risk factors and risk estimates. After careful consideration we decided to leave the table in the supplemental material due to its big size, which makes it a bit clumsy to read. We hope that the supplemental material will be easily accesible to the reader.

  1. Table 2 is not crear. Please indicate what represents the numbers under "risk factors"  

Thank you for pointing out that the table is not clear. We revised Table 2 for clarity (line 197 in the revised manuscript) with clearer indication of units and column headings.

Reviewer 3 Report

This is a very interesting and novel paper. I have a few minor comments, mostly on the Materials and Methods section, as follows:

Abstract: lines 23 and 26/7 - 'childhood overweight' could be replaced with 'being overweight in childhood' and then your sentence in lines 26/7 would make better grammatical sense. 

Abstract: line 28/9 - "The results confirm the potential to explain additional part of the non-communicable diseases by the DOHAD concept...' - 'additional part' doesn't quite fit here - I would suggest rewording this sentence. 

I appreciate that the supplementary file contains further details of the literature review. However I think it would be beneficial for the reader to include the following detail in the main text as additional information:

  • Were there any restrictions on literature included in the review by year of publication? (ie papers published within the last 10 years?)
  • Why was only PubMed searched and no other databases included?
  • When were the searches undertaken? 

Line 150: 'Statistically significant association for smoking during pregnancy was reported only low birth weight.' This sentence doesn't make sense as is and should be reviewed accordingly.

Line 208: Remove the comma after 'cardiovascular diseases'.

Author Response

Dear Reviewer,

Thank you for your time and effort and the highly useful comments. We have taken them into account and adjusted the manuscript accordingly. Please find below our detailed responses. Additions and changes are marked in the manuscript with Word Track changes.

This is a very interesting and novel paper. I have a few minor comments, mostly on the Materials and Methods section, as follows:

  1. Abstract: lines 23 and 26/7 - 'childhood overweight' could be replaced with 'being overweight in childhood' and then your sentence in lines 26/7 would make better grammatical sense.

Thank you for pointing out that the sentence is not correct and presenting a suggestion to improve it. According to it, we revised the sentence. Please see lines 26f. in the revised manuscript.

  1. Abstract: line 28/9 - "The results confirm the potential to explain additional part of the non-communicable diseases by the DOHAD concept...' - 'additional part' doesn't quite fit here - I would suggest rewording this sentence.

Thank you for pointing out that the sentence was not clear. We revised it to make it understandable and grammatically correct. Please see line 29 in the revised manuscript.

  1. I appreciate that the supplementary file contains further details of the literature review. However I think it would be beneficial for the reader to include the following detail in the main text as additional information:
  • Were there any restrictions on literature included in the review by year of publication? (ie papers published within the last 10 years?)
  • Why was only PubMed searched and no other databases included?
  • When were the searches undertaken? 

We appreciate your suggestions to improve the understandability of our manuscript. We expanded the description of the literature review according to your comments (lines 116ff. in the revised manuscript). This work was a scoping attempt to test whether an approach, as the one presented in the manuscript, is feasible. Thus, we designed the work in a way that it also includes only scoping literature review, i.e. limited to published meta-analyses and searches only in PubMed. We agree that future work should include a more comprehensive literature search, i.e. in several databases and of original studies, if needed supplemented with meta-analysis.

  1. Line 150: 'Statistically significant association for smoking during pregnancy was reported only low birth weight.' This sentence doesn't make sense as is and should be reviewed accordingly.

Thank you for pointing out that the sentence did not make sense. The sentence has been revised; see line 160 in the revised manuscript.

  1. Line 208: Remove the comma after 'cardiovascular diseases'.

Thank you for highlighting this error. We removed the comma; see line 217 in the revised manuscript.